# Changes in physical activity, dietary and sleeping pattern among the general population in COVID-19: A systematic review protocol

**Pa Pa Soe** [1]*, **Zar Lwin Hnin**[1◉], **Thein Hlaing**[2◉], **Hlaing Min**[1]

1 Department of Preventive and Social Medicine, University of Medicine, Mandalay, Mandalay, Myanmar,
2 District Public Health Department, Pyay District, Bago Region, Myanmar

◉ These authors contributed equally to this work.
* papasoe.jpn@gmail.com

## Abstract

**Data Availability Statement:** No datasets were generated or analysed during the current study. All relevant data from this study will be made available upon study completion.

### Background

COVID-19 is a highly infectious respiratory disease caused by a new coronavirus known as SARS-CoV-2. Home confinement and movement restrictions can affect lifestyle changes and may lead to non-communicable diseases (NCD). This systematic review will provide a detailed summary of changing patterns of physical activities, diet and sleep among the general public in COVID-19.

### Methods

PubMed, Google Scholar, EMBASE, Science Direct, and Scopus will be, among eight bibliographic databases, applied and search work will take one month (from January 2021 until February 2021). Key search terms will include common characteristics of physical activity, dietary pattern, sleeping pattern, and COVID-19. The reviewers will fully apply the inclusion and exclusion criteria framed by PICOS as well as the screening form and the PRISMA flow for selecting the papers eligible for this review. Moreover, the reviewers will use a self-developed excel table to extract the required information on dietary pattern changes, physical activities and sleep patterns changes, and the Risk of Bias Assessment Tool for Nonrandomized Studies (RoBANS) for practicing quality assessment. We will include only observational studies and analyze the extracted information qualitatively and the review output will be based on the eligible studies' outcomes.

### Discussion

Changes in physical activity, dietary and sleep patterns are challenging to the public health professionals regarding the risk factors for NCD, and long-term effects might impact the controlling of the NCD. Evidence-based research information is needed regarding the COVID-19 pandemic, and there are a few global data on changes in physical activity, dietary and sleep patterns. Furthermore, innovative public health interventions or implementations are

**Funding:** The author(s) received no specific funding for this work.

**Competing interests:** The authors have declared that no competing interests exist.

needed to maintain the positive health status of the population in the long run as the consequences of the COVID-19 pandemic.

## Systematic review registration

This systematic review is based on a protocol registered with PROSPERO CRD42021232667.

## Introduction

The coronavirus disease 2019 (COVID-19), developed by the severe acute respiratory syndrome coronavirus 2 (SARS-CoV-2) that can transform the different forms of variants: Alpha-B.1.1.7, Beta-B.1.351, Gamma-P.1, Delta-B.1.617.2 and Omicron-BA.1 or BA.2 [1]. Its infectivity has speeded into a worldwide pandemic, causing not only health issues (physical and psychological issues) but also devastating economic conditions [2–4]. Public health professionals implemented an epidemiological containment strategy to control the COVID-19 by using lockdown measures [5] since lifting the strategic restrictions has some important advantages such as limiting the spread of the COVID-19 virus, enhancing more preparation time for the preventive strategies and avoiding the new peak of the COVID-19 outbreak [6]. However, these restriction measures are mandatory and therefore have a strong negative impact on healthy lifestyle behaviors such as physical activity (PA), eating habits, quality of sleep, etc. [7–12]. Hence, although individuals were encouraged to remain physically active in their homes, the unprecedented confinement gave rise to two situations. (i) the physically active people decreased their activity, and (ii) the inactive population were not likely to increase their daily PA [10, 13]. For instance, the prolonged homestay decreased the amount of daily PA performed [14]. Similarly, it has been suggested that due to this period of abruptly reduced PA, changes in eating patterns, such as overeating, will start to emerge change in dietary patterns, putting people at risk for revealing an eating disorder [15]. This lack of PA due to home confinement has also been a potential risk factor that negatively affects sleep quality [5, 16].

The recent reviews demonstrated that acute and regular PA improved sleep quality. PA improved sleep even in patients suffering from insomnia or sleep apnea. Diet is also widely considered an important modifiable factor that has been often proposed to improve sleep duration and quality [17, 18]. Evidence from a multi-continental survey among adults indicated that during the COVID-19 pandemic, the frequency and duration of PA decreased by 24% and 33.5%, respectively. The study stated significant increases in consumption of unhealthy food, eating out of control, and snacking between meals [19]. The studies revealed that PA decreased although eating pattern was healthy [11] while those who adopted unhealthy lifestyle encountered the lower sleep quality [12]. Regarding sleep quality, 17.2% answered that sleep quality improved, 56.5% did not change, and 26.3% worsened compared to a typical week [17]. The confinement had a significant differential effect on physically active participants, who experienced a substantial decline in their PA levels, sleep quality, and well-being [5]. Surveillance of the lockdown results on healthy habits should become a routine as part of readiness endeavors worldwide [20].

Due to the COVID-19 confinements, almost all areas of public life have been affected and changed. Many lives across the globe experience a wide range of changes that may be intended or unintended as well as positive or negative. For instance, many people become familiar with the words or terminologies of an epidemic disease like "social distancing", "quarantine", "stay

home", and "new normal" and healthy practices like "wearing a mask", and "washing hands". Inversely, the COVID-19 pandemic carries unintended or negative changes in the proportions of educational achievements, employments, individual net income, supply chain quality, movements and food and drug availability [21]. Changes in PA, dietary and sleep patterns are challenging to the public health professionals regarding the risk factors for further illnesses especially **non-communicable diseases** (NCD), and long-term effects might impact the controlling of the NCD. Currently, evidence-based research information is needed regarding the COVID-19 pandemic, and there are a few global data on changes in physical activity, dietary and sleep patterns. The analysis of this review focuses on the changing patterns of PA, diet and sleep since the COCID-19 restrictions were introduced and intends to conclude which changes are positive or negative as well as healthy or unhealthy. Especially, the interest of this review is to provide reliable and collective evidence to inform the important interventions for how to adjust the peoples' changes in their PA, dietary and sleeping patterns in order for mitigating the risks of further illnesses.

## Materials and methods

Firstly, we screened the International Prospective Register of Systematic Reviews database [PROSPERO], ensuring previous systematic reviews or meta-analyses have not been taken to this research question. This protocol was registered and published in PROSPERO on January 8 2021 [ID: CRD42021232667] to provide the transparency of the science. Furthermore, to guarantee the straightforward and total endeavor, maximize the benefits, and to be a compelling course of action of announcing of this review, the PRISMA-P [Preferred Reporting Items for Systematic Reviews and Meta-Analyses Protocol] 2015 Checklist (S1 File) will be totally applied.

### Eligibility criteria according to PICOS

Applied eligibility criteria are enumerated below according to the PICOS scheme. The review question formulated with the PICOS frame is "Due to the COVID-19 restrictions, what changing patterns ofPA, diet and sleep will occur among the general population after October 2019".

   **Participants.**   All trials on people ($\geq$ 12 years) of either sex will be eligible for inclusion with the following exceptions. The studies which worked on the healthcare workers and the patients with any diseases will be excluded.

   **Interventions.**   Eligible studies must investigate the effects of COVID-19 safety measures. It was defined as any public measure implemented to halt the transmission of COVID-19, lockdown measure including physical and social distancing, self-isolation, closure of public facilities, restriction of public events and gatherings, travel limitation, and working from home [4, 21].

   **Controls.**   This review will not try to analyze the statistical significances of the changing effects ofPA, diet and sleep between healthy and unhealthy groups. Therefore, control studies and comparisons between general population and other control groups will not be considered to include in this review.

   **Outcomes.**   Changes in PA, sleep patterns and diet patterns must be assessed as the study outcomes. PA was defined as any bodily movement by the skeletal musculature resulting in energy expenditure [22]. Indeed, the most recent WHO guidelines on physical activity [23] explicitly stated that every minute of PA counts. The studies will be considered for their eligibility when sleep was described by better, unchanged and worsen sleep quality as well as when measuring types and frequencies of food intakes and examining body weight status in their reports. This review aims to present the results of the outcomes by categorical data (increased/

decreased/unchanged). The assessment can be done via (online) questionnaires, interviews but is not limited to these.

**List and definitions of variables.**

1. Population–Study participants (≥12years) of both sexes. Healthcare workers and patients with any diseases will be excluded.

2. COVID-19 safety measures–It was defined as any lockdown measures to cut the transmission of the COVID-19, for example, closure of the public facilities, travel restriction, social distancing and work from home.

3. Changes–will be defined as increased/decreased/unchanged interest variables.

4. PA–was defined as the following conditions. Indoor or/and outdoor PA, PA with some types of body movement exercises, PA through leisure activities, intensity of the PA (vigorous, moderate, low), duration of the PA (shortened/lengthened/unchanged), mean minutes of the PA. And changes in time, frequency and intensity of PA will be reviewed as the outcome.

5. Dietary pattern–was defined as the following: eating behavior of healthy/unhealthy diet, consumption of vegetables and fruits, consumption of fast food, junk food and fried food, skipping meals.

6. Sleep pattern—was defined as the following: sleeping time (shorter/longer/unchanged), mean sleep duration (hours/minutes), timing of the sleep (early/late), sleep quality (poor/improved/unchanged).

7. BMI–The results of the BMI in the reviewed articles will be used and categorized into groups.

**Study design.**   The primary researches, including observational studies (longitudinal and cross-sectional) in English, will be included. Experimental trials applying PA, duplicated, or abstract-only papers, case reports and case series, editorials, and documents with no full-text available will be excluded. Especially, the eligible studies have to be designed to focus on the effects of preventive public health measures in the context of the COVID-19 pandemic. Finally, only studies published after October 2019 will be eligible for inclusion because this review wants to explore the events of many countries affected by the COVID-19 and the COVID-19 widely spreads across the world after October 2019.

## Information sources and search strategy

Optimal searches in this review will be exercised via PubMed, Google Scholar, EMBASE, Science Direct, and Scopus. The data search strategy provides the search method for at least one electronic database, including planning constraints. In this study, searches for articles or journals using keywords and Boolean's Operator [AND, OR, NOT and NEST or brackets and quotation marks] will be used to expand or narrow the search so that it makes it easier to determine which articles or journals are used and take from MeSH [Medical Subject Headings] on PubMed which can be seen in the Table 1. Developing the screening form (including search terms, number of references screened per time, per reviewer and per database, reasons for inclusion/exclusion), exercising independent searches, discussions among reviewers will be ensured. Besides, inter-reviewer agreement form (including "Paper Code", "Paper Title", "Includable-Yes", "Excludable-No", Uncertain-Maybe" and "Reasons for Inclusion/Exclusion") will be developed and practiced for systematic selection of the studies. The completed search

**Table 1. Search strategy.**

| COVID-19 | Physical Activity | Dietary | Sleeping |
|---|---|---|---|
| 2019-nCOV | Duration (PA) | Diet pattern | Sleeping pattern |
| OR | | OR | OR |
| SARS-CoV-2 | Frequency (PA) | Eating habits | Sleeping disturbances |
| OR | | OR | OR |
| Corona Virus | Intensity (PA) | Healthy and Unhealthy food | Sleeping disorders |
| OR | | | OR |
| COVID-19 | Types (PA) | | Sleep quality |

will take about one month and therefore will be planned to start on January 8, 2021, and end on February 8, 2021. PRISMA 2020 flow diagram for new systematic reviews which included searches of databases and registers only was applied in the protocol (S2 File).

## Data extraction and management

The search results from all sources of evidence will be documented in the Mendeley library for preventing subsequent duplication of article data extraction. The study selection process will be based on the PRISMA flow diagram. At least two reviewers will carry out the screening process based on the eligibility criteria by reading the title and abstract. Secondly, reference lists from the literature will be identified in the evidence search stage. Furthermore, screening was carried out based on the eligibility criteria by reading the literature's full text, which passed the title and abstract screening stages. Two reviewers will independently extract and manage data for each included study using self-developed excel table. Data will be extracted according to the inclusion and exclusion criteria based on the PICOS eligibility criteria, the collection of information on dietary pattern changes and lifestyle changes, and the quality assessment report for each article. In this stage, we will discuss the opinions regarding study population, study design, outcomes and measurements of the outcomes of the studies. The differences of opinions between the two reviewers will be resolved using discussion. If no agreement is reached, consultations will be conducted with third parties not involved in the selection and extraction processes.

## Risk of bias (quality) assessment

Risk of Bias Assessment Tool for Nonrandomized Studies (RoBANS) [24] will be used. Two reviewers will independently carry out the review process to assess the risk of bias for each included study. If any disagreements exit between the reviewers, the third reviewer will be assessed the risk. Each outcome within the studies will be graded as having a high, low or unclear risk of bias.

## Strategy for data synthesis

This research uses descriptive analysis. The extracted information will be qualitatively analyzed, and the review output will base on variability in outcomes of the eligible studies. The analysis method with stages is as follows:

i. Summarizing the included studies' characteristics according to the study designs, sample sizes, study focus areas and objectives.

ii. Summarizing all extracted information by grouping it into predetermined themes and analyzing them in term of the qualitative synthesis.

### Ethical considerations

**Research ethics approval.** Not applicable. This review is based on secondary analyses of public information recognized through a systematic search and review process.

## Discussion

### Limitations of the study design

We will only review the observational studies and it may introduce many biases (selection bias and recall bias). And then causal inference cannot be concluded. Most of the studies collected self-reported data on all outcome's variables. Moreover, measurements of the PA, and sleep quality were subjective and dietary intake introduced the recall bias. Therefore, accurate data on the changes of PA, sleep pattern and diet pattern might not be presented.

### Access to data

The taking after data will be made freely accessible for all included data sources: reference information or contact name/institution, population represented, data collection method, questionnaire used, and sample size as relevant.

## Supporting information

**S1 File. PRISMA-P (Preferred Reporting Items for Systematic Review and Meta-analysis Protocols) 2015 checklist.**
(DOC)

**S2 File. PRISMA 2020 flow diagram for new systematic reviews which included searches of databases and registers only.**
(TIF)

## Acknowledgments

We are grateful to Mrs. Moe Shwe Sin Chit for giving your time in copy editing of the manuscript.

## Author Contributions

**Conceptualization:** Pa Pa Soe, Hlaing Min.

**Methodology:** Pa Pa Soe, Zar Lwin Hnin, Thein Hlaing.

**Writing – original draft:** Pa Pa Soe, Zar Lwin Hnin.

**Writing – review & editing:** Pa Pa Soe, Zar Lwin Hnin, Thein Hlaing.

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
