## [Decision Letter · Decision Letter 0]

14 Feb 2022

PONE-D-21-40766Changes in physical activity, dietary and sleeping pattern among the general population in COVID-19 : A systematic review protocolPLOS ON

Thank you for submitting your manuscript to PLOS ONE. After careful consideration, we feel that it has merit but does not fully meet PLOS ONE’s publication criteria as it currently stands. Therefore, we invite you to submit a revised version of the manuscript that addresses the points raised during the review process.

We look forward to receiving your revised manuscript.

Kind regards,

Luigi Lavorgna

Academic Editor

PLOS ONE

Journal Requirements:

3. We note that this manuscript is a systematic review or meta-analysis; our author guidelines therefore require that you use PRISMA guidance to help improve reporting quality of this type of study. Please upload copies of the completed PRISMA checklist as Supporting Information with a file name “PRISMA checklist”.

Reviewers' comments:

Reviewer's Responses to Questions

**Comments to the Author**

1. Does the manuscript provide a valid rationale for the proposed study, with clearly identified and justified research questions?

Reviewer #1: Yes

2. Is the protocol technically sound and planned in a manner that will lead to a meaningful outcome and allow testing the stated hypotheses?

Reviewer #1: Partly

3. Is the methodology feasible and described in sufficient detail to allow the work to be replicable?

Reviewer #1: Yes

4. Have the authors described where all data underlying the findings will be made available when the study is complete?

Reviewer #1: Yes

5. Is the manuscript presented in an intelligible fashion and written in standard English?

Reviewer #1: Yes

6. Review Comments to the Author

You may also provide optional suggestions and comments to authors that they might find helpful in planning their study.

Reviewer #1: In this protocol proposal, Pa Pa Soe et al describe a possible review protocol to investigate how COVID-19-related restrictions impacted on sleep quality, physical exercise and dietary pattern.

The aim is quite ambitious, as the amount of manuscripts on the topic has steeply surged within the last year in relation with both healthy conditions and diseases. I would better highlight why authors decided to only include manuscript published between January and February 2021 and also I would further introduce the possible association between the physical exercise and quality of life, including the sleeping status, in both healthy controls and in patients with other diseases (see this works Durucan et al. 2022 [10.1097/MRR.0000000000000519]; Cunha et al. 2021 [10.1016/j.numecd.2021.12.019]; Carotenuto et al. 2021 [10.3390/jcm10061234]; Beydoun et al. 2022 [10.1093/gerona/glac028]; Bruno et al. 2022 [10.1016/j.sleep.2022.01.002]). All the aforementioned manuscript usually referred to people’s condition during the period you choose for selecting manuscript and hence, during the more severe COVID 19 restrictions.

7. PLOS authors have the option to publish the peer review history of their article (what does this mean?). If published, this will include your full peer review and any attached files.

Reviewer #1: No

---

## [Author Response · Author response to Decision Letter 0]

30 Mar 2022

List of Amendments 

Line number- 24 (Abstract)

"Were" was corrected by "will be" because it should be future tense. 

Line number- 25 (Abstract)

"applied and search work will take one month" were added to clearly mention the planned time of search work. 

Line number- 26 (Abstract)

"usages" was substituted by "characteristics" because key search terms include the characteristics of physical activity (e.g. frequency, duration), diet (e.g. patterns, food types) and sleep (quality, timing). 

Line number- 28 and 29 (Abstract)

" as well as the screening form and the PRISMA flow" were added because these are important information for the selection process of this review. 

Line number-55 (Introduction)

Citation [5] was moved to the end of the relevant information in line numbers- 53 to 55 and put together with citations [6, 7, 8] (Line number-55) because these discussion points were concerned with citation [5]. 

Line number- 58 (Introduction)

Citation [8] was moved to the end of the relevant information and put together with citation [9] (Line number-58) because citation [8] covered the information of line numbers- 55 to 58.

Line number- 67 (Introduction)

Citations [13, 14] were removed due to duplication and putting them [13,14] (Line number-67) at the end of the summarized or paraphrased information will be more relevant.

Line number- 69 (Introduction)

Citation [15] was removed due to duplication.

Line number- 75 (Introduction)

Citation [4] was removed because the suggestion points cited with [4] and [16] were more relevant to only citation [16]. 

Line number- 76 (Last paragraph of the Introduction Section)

The last paragraph of the introduction section was substituted by some information in order to be the relevant and strong justification of the review objective. 

Line number- 99 (Methods) 

"a" was added for grammar correctness.

Line number- 102 (Methods)

"will be totally applied" were added for grammar correctness.

Line number- 105 (Eligibility Criteria)

"the effects of preventive public health measures" were substituted by "the Covid-19 restrictions" which are more precise words. 

Line number- 118-121 (Controls)

The paragraph under the sub-title of "Controls" was removed and restructured with " This review will not try to analyze the statistical significances of the changing effects of physical activity, diet and sleep between healthy and unhealthy groups. Therefore, control studies and comparisons between the general population and other control groups will not be considered to include in this review." 

Line number- 133 (List and Definitions of Variables)

"sex" was corrected by "sexes" for grammar correctness. 

Line number- 158-160 (Study Design)

"because this review wants to explore the events of many countries affected by the Covid-19 and the Covid-19 widely spread across the world after October 2019" were added for justifying the publication date eligible for this review. 

Line number- 168-175 (Information Sources and Search Strategy)

"Developing the screening form (including search terms, number of references screened per time, per reviewer and per database, reasons for inclusion/exclusion), exercising independent searches, discussions among reviewers will be ensured. Besides, inter-reviewer agreement form (including "Paper Code", "Paper Title", "Includable-Yes", "Excludable-No", Uncertain-Maybe" and "Reasons for Inclusion/Exclusion") will be developed and practiced for systematic selection of the studies. The completed search will take about one month and therefore will be planned to start on January 8, 2021, and end on February 8, 2021." were added for providing more information of search and selection processes. 

Line number-179 (Table.1. Search Strategy)

"Duration (PA), Frequency (PA), Intensity (PA) and Types (PA)" were added under the column of Physical Activity for application of detailed search.

Line number- 216 (Limitations of the study design)

"bias" was corrected by "biases" for grammar correctness. 

PRISMA checklist

In the connection with previous email, we uploaded the “PRISMA checklist” in this time. However, we had already uploaded the PRISMA-P checklist (S1 file) and I think it is entitled for the study protocol. May I upload both files if you don’t mind.

---

## [Editor Report · Decision Letter 1]

1 Apr 2022

PONE-D-21-40766R1Changes in physical activity, dietary and sleeping pattern among the general population in COVID-19 : A systematic review protocolPLOS ONE

Dear Dr. Soe,

Thank you for submitting your manuscript to PLOS ONE. After careful consideration, we feel that it has merit but does not fully meet PLOS ONE’s publication criteria as it currently stands. Therefore, we invite you to submit a revised version of the manuscript that addresses the points raised during the review process.

Please include all references that the auditor has indicated to you.

We look forward to receiving your revised manuscript.

Kind regards,

Luigi Lavorgna

Academic Editor

PLOS ONE 
---

## [Author Response · Author response to Decision Letter 1]

12 May 2022

List of Amendments

1. Line 49-51 (Introduction)

“that can transform the different forms of variants: Alpha-B.1.1.7, Beta-B.1.351, Gamma-P.1, Delta-B.1.617.2 and Omicron-BA.1 or BA.2 [1]. Its infectivity“ was added and Citation [1] was newly added.

2. Line 52 (Introduction)

Citation number was changed from [1-3] to [2-4].

3. Line 54 (Introduction)

Citation number was changed from [4] to [5].

4. Line 54-57 (Introduction)

"This mandatory self-isolation may affect people’s physical and mental health" were was substituted by "since lifting the strategic restrictions has some important advantages such as limiting the spread of the Covid-19 virus, enhancing more preparation time for the preventive strategies and avoiding the new peak of the Covid019 outbreak [6]. However, these restriction measures are mandatory". Citation number [6] was newly added.

5. Line 59 (Introduction) (Reviewer #1’s comment)

Citation number was changed from [5-8] to [7-12] and citation number [11] and [12] was newly added to provide evidence for relevant information.

6. Line number- 60 (Introduction)

"may give rise" was changed to "gave rise" for correction. 

7. Line number- 61 (Introduction)

"the physically active people may decrease their activity" was changed to "the physically active people decreased their activity " for correction. 

8. Line number- 61-62 (Introduction)

"may not be likely to increase their daily PA" was changed to "were not likely to increase their daily PA" for correction.

9. Line number- 62 (Introduction)

"prolonged homestay may decrease the amount of daily PA performed" was changed to " prolonged homestay decreased the amount of daily PA performed " for correction.

10. Line 62 (Introduction)

Citation number was changed from [8,9] to [10,13].

11. Line 63 (Introduction)

Citation number was changed from [10] to [14].

12. Line 65 (Introduction)

Citation number was changed from [11] to [15].

13. Line 67 (Introduction)

Citation number was changed from [4,12] to [5,16].

14. Line 71 (Introduction)

Citation number was changed from [13,14] to [17,18].

15. Line 74 (Introduction)

Citation number was changed from [15] to [19].

16. Line 74 - 76 (Introduction) (Reviewer #1’s comment)

‘The studies revealed that PA decreased although eating pattern was healthy [11] while those who adopted unhealthy lifestyle encountered the lower sleep quality [12].’ was newly added and the citation number [11] and [12] was also newly cited. 

17. Line 78 (Introduction)

Citation number was changed from [13] to [17].

18. Line 80 (Introduction)

Citation number was changed from [4] to [5].

19. Line 81 (Introduction)

Citation number was changed from [16] to [20].

20. Line 89 (Introduction)

Citation number was changed from [17] to [21].

21. Line 122 (Materials and Methods)

Citation number was changed from [3,17] to [4,21].

22. Line 131 (Materials and Methods)

Citation number was changed from [18] to [22].

23. Line 131 (Materials and Methods)

Citation number was changed from [19] to [23].

24. Line 204 (Risk of bias (quality) assessment)

Citation number was changed from [20] to [24].

25. Line number- 248-249 (References)

The following reference was added.

1. Genomic Epidemiology of hCov-19 [Internet]. GISAID - Initiative. 2022 [cited 2022 Apr 30]. Available from: https://www.gisaid.org/

26. Line number- 263-266 (References)

The following reference was added.

6. Coronavirus disease (COVID-19): Herd immunity, lockdowns and COVID-19 [Internet]. WHO | World Health Organization. [cited 2022 Apr 30]. Available from: https://www.who.int/news-room/questions-and-answers/item/herd-immunity-lockdowns-and-covid-19

27. According to reviewer #1, Durucan et al. 2022 [10.1097/MRR.0000000000000519]; Cunha et al. 2021 [10.1016/j.numecd.2021.12.019]; Carotenuto et al. 2021 [10.3390/jcm10061234]; Beydoun et al. 2022 [10.1093/gerona/glac028]; Bruno et al. 2022 [10.1016/j.sleep.2022.01.002] were suggested to investigate how COVID-19-related restrictions impacted on sleep quality, physical exercise and dietary pattern. 

We had cited both Cunha et al. 2021 [10.1016/j.numecd.2021.12.019] and Bruno et al. 2022 [10.1016/j.sleep.2022.01.002] in the manuscript, however, the rest articles were left with the reason of unaligned study population (patients).

---

## [Editor Report · Decision Letter 2]

17 May 2022

Changes in physical activity, dietary and sleeping pattern among the general population in COVID-19 : A systematic review protocol

PONE-D-21-40766R2

We’re pleased to inform you that your manuscript has been judged scientifically suitable for publication and will be formally accepted for publication once it meets all outstanding technical requirements.

Kind regards,

Luigi Lavorgna

Academic Editor

PLOS ONE
---

## [Editor Report · Acceptance letter]

25 May 2022

PONE-D-21-40766R2 

Changes in physical activity, dietary and sleeping pattern among the general population in COVID-19: A systematic review protocol 

Dear Dr. Soe:

I'm pleased to inform you that your manuscript has been deemed suitable for publication in PLOS ONE. Congratulations! Your manuscript is now with our production department. 

Kind regards, 

on behalf of

Dr. Luigi Lavorgna 

Academic Editor

PLOS ONE